# Semantically Adversarial Scene Generation with Explicit Knowledge Guidance for Autonomous Driving

## Abstract

Generating adversarial scenes that potentially fail autonomous driving systems provides an effective way to improve their robustness. Extending purely data-driven generative models, recent specialized models satisfy additional controllable requirements such as embedding a traffic sign in a driving scene by manipulating patterns *implicitly* at the neuron level. In this paper, we introduce a method to incorporate domain knowledge *explicitly* in the generation process to achieve *Semantically Adversarial Generation (SAG)*. To be consistent with the composition of driving scenes, we first categorize the knowledge into two types, the property of objects and the relationship among objects. We then propose a tree-structured variational auto-encoder (T-VAE) to learn hierarchical scene representation. By imposing semantic rules on the properties of nodes and edges into the tree structure, explicit knowledge integration enables controllable generation. To demonstrate the advantage of structural representation, we construct a synthetic example to illustrate the controllability and explainability of our method in a succinct setting. We further extend to realistic environments for autonomous vehicles, showing that our method efficiently identifies adversarial driving scenes against different state-of-the-art 3D point cloud segmentation models and satisfies the traffic rules specified as explicit knowledge.

## 1 Introduction

According to the report published by the California Department of Motor Vehicle (DMV, 2022), there were at least five companies (Waymo, Cruise, AutoX, Pony.AI, Argo.AI) that made their autonomous vehicles (AVs) drive more than 10,000 miles without disengagement. It is a great achievement that current AVs succeed in normal cases trained by hundreds of millions of miles of training. However, we are still unsure about their safety and robustness in rare but critical driving scenes, e.g., the perception system fails to detect a pedestrian that is partially blocked by a surrounding vehicle. One promising solution could be artificially generating driving scenes in simulations to find potential failures of AV systems. The biggest difficulty of creating such adversarial scenes is incorporating traffic rules and semantic knowledge to make the generation realistic and controllable.

The recent breakthrough in machine learning enables us to learn complex distributions with sophisticated models, which uncover the data generation process so as to achieve controllable data generation (Abdal et al., 2019; Tripp et al., 2020; Ding et al., 2021). Deep Generative Models (DGMs) (Goodfellow et al., 2014; Kingma & Welling, 2013), approximating the data distribution with neural networks (NN), are representative methods to generate data targeting a specific style or category. However, existing controllable generative models focus on manipulating implicit patterns at the neuron or feature level. For instance, Bau et al. (2020) dissects DGMs to build the relationship between neurons and generated data, while Plumerault et al. (2020) interpolates in the latent space to obtain vectors that control the poses of objects. One main limitation is that they cannot explicitly incorporate semantic rules, e.g., cars follow the direction of lanes, which may lead to meaningless data that violates common sense. In light of the limitation, we aim to develop a structural generative framework to integrate explicit knowledge (Dienes & Perner, 1999) during the generation process and thus control the generated driving scene to be compliant with semantic rules.

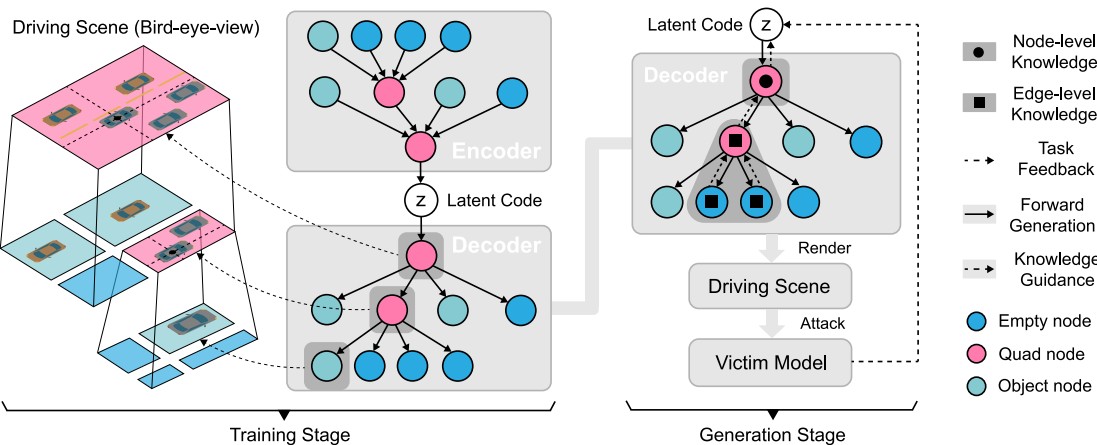

Figure 1: Diagram of SAG. **Training stage.** Train the tree generative model to learn the representation of structured data. **Generation stage.** Integrate node-level and edge-level knowledge during the generation to create adversarial samples for the victim model.

Driving scenes can be described with objects and their various relationships (Amizadeh et al., 2020). Thus, in this paper, we categorize the semantic knowledge that helps scene generation into two types. The first type denoted as *node-level knowledge* represents the properties of single objects and the second type denoted as *edge-level knowledge* represents the relationship among objects. We observe that tree structure is highly consistent with this categorization for constructing scenes, where nodes of the tree represent objects and edges the relationship. We can explicitly integrate the *node-level* and *edge-level* knowledge by manipulating the tree structure during the generation.

In this paper, we propose the framework Semantically Adversarial Generation (SAG) as shown in Figure 1. SAG contains two stages to separate the learning of data distribution of real-world driving scenes and the searching of adversarial scenes with knowledge as constraints. In the *training stage*, we train a tree-structured generative model that parameterizes nodes and edges of trees with NN to learn the representation of structured data. In the *generation stage*, explicit knowledge is applied to different levels of the learned tree model to achieve knowledge-guided generation for reducing the performance of victim algorithms.

To verify our method, we first construct a synthetic reconstruction example to illustrate its advantages and provide an analysis of its controllability and explainability. With SAG, it is possible to generate natural scenes that follow semantic rules, e.g., boxes with the same color should be positioned close to each other. To demonstrate the practicality of SAG, we conduct extensive experiments on adversarial LiDAR scene generation against 3D segmentation models. We show that our generated driving scenes successfully attack victim models and meanwhile follow the specified traffic rules. In addition, compared with traditional attack methods, scenes generated by our method achieve stronger adversarial transferability across different victim models. Our technical contributions are summarized below:

- We propose a semantically adversarial generative framework (SAG) via integrating explicit knowledge and categorizing the knowledge into two types according to the composition of driving scenes.

- We propose a tree-structured generative model based on our knowledge categorization and construct a synthetic example to demonstrate the effectiveness of our knowledge integration.

- We propose Scene Attack, the *first* semantic adversarial point cloud attack based on SAG, against state-of-the-art segmentation algorithms, which demonstrates several essential properties.

## 2 Related Work

**Semantically adversarial attacks.** Traditional adversarial attack methods focused on the pixel-wise attack in the image field, where $L_p$-norm is used to constrain the adversarial perturbation. For the sake of

the interpretability of adversarial samples, recent studies begin to consider *semantic attacks*. They attack the rendering process of images by modifying the light condition (Liu et al., 2018; Zeng et al., 2019) or manipulating the position and shape of objects (Alcorn et al., 2019; Xiao et al., 2019; Jain et al., 2019). This paper explores the generation of adversarial point cloud scenes, which already have similar prior works (Tu et al., 2020; Abdelfattah et al., 2021; Sun et al., 2020a). Tu et al. (2020) and Abdelfattah et al. (2021) modify the environment by adding objects on top of existing vehicles to make them disappear. Sun et al. (2020a) create a ghost vehicle by adding an ignorable number of points; however, they modify a single object without considering the structural relationship of the whole scene.

**Semantic driving scene generation.** Existing ways of scene generation focus on sampling from pre-defined rules and grammars, such as probabilistic scene graphs used in Prakash et al. (2019) and heuristic rules applied in Dosovitskiy et al. (2017). These methods rely on domain expertise and cannot be easily extensible to large-scale scenes. Recently, data-driven generative models (Devaranjan et al., 2020; Tan et al., 2021; Para et al., 2020; Li et al., 2019; Kundu et al., 2018) are proposed to learn the distribution of objects and decouple the generation of scenes into sequence (Tan et al., 2021) or graphs (Para et al., 2020; Li et al., 2019). Although they reduce the gap between simulation and reality, generated scenes cannot satisfy specific constraints. Another substantial body of literature (Eslami et al., 2016; Kosiorek et al., 2018; Gu et al., 2019) explores directly learning scene graphs from images via an end-to-end framework. Their generalization to high-dimensional data is very challenging, making them less effective than modularized methods proposed by Kundu et al. (2018); Wu et al. (2017); Devaranjan et al. (2020).

**Structural deep generative models.** Most of DGMs, such as Generative Adversarial Networks (GAN) (Goodfellow et al., 2014) and Variational Auto-encoder (VAE) (Kingma & Welling, 2013), are used for unstructured data. They leverage the powerful feature extraction of NNs to achieve impressive results (Karras et al., 2019; Brock et al., 2018). However, the physical world is complex due to the diverse and structural relationships of objects. Domain-specific structural generative models are developed via tree structure e.g., RvNN-VAE (Li et al., 2019) or graph structure e.g., Graph-VAE (Simonovsky & Komodakis, 2018). Rule-based generative models are also explored by sampling from pre-defined rules (Kusner et al., 2017; Kar et al., 2019; Devaranjan et al., 2020). One practical application of structural DGMs is generating samples to satisfy requirements of downstream tasks (Engel et al., 2017; Tripp et al., 2020). Abdal et al. (2019) and Abdal et al. (2020) search in the latent space of StyleGAN (Karras et al., 2019) to obtain images that are similar to a given image. For structured data, such a searching framework transforms discrete space optimization to continuous space optimization, which was shown to be more efficient (Luo et al., 2018). However, it may not guarantee the rationality of generated structured data due to the loss of interpretability and constraints in the latent space (Dai et al., 2018).

**Incorporating knowledge into neural networks.** Integrating knowledge into data-driven models has been explored in various forms from training methods, meta-modeling, and embedding to rules used for reasoning. (Hu et al., 2016) distills logical rules with a teacher-student framework under *Posterior Regularization* (Ganchev et al., 2010). Another way of knowledge distillation is encoding knowledge into vectors and then refining the features from the model that are in line with the encoded knowledge (Gu et al., 2019). These methods need to access *Knowledge Graphs* (Ehrlinger & Wöß, 2016) during the training, which heavily depends on human experts. Meta-modeling of complex fluid is integrated into the NN to improve the performance of purely data-driven networks in (Mahmoudabadbozchelou et al., 2021). In addition, (Yang & Perdikaris, 2018) restricts the output of generative models to satisfy physical laws expressed by partial differential equations. In the reinforcement learning area, reward shaping (Ng et al., 1999) is also recognized as one technique to incorporate heuristic knowledge to guide the training.

## 3 Semantically Adversarial Generation Framework

We define the driving scene $x \in \mathcal{X}$ in the physical world, which contains a group of objects and their properties such as positions and colors. The goal of our framework is to generate $x$ so that to reduce the performance $\mathcal{L}_t(x)$ of the victim model $t \in \mathcal{T}$, as well as satisfying semantic loss $\mathcal{L}_\mathbb{K}(x)$:

$$x = \arg\min_x \mathcal{L}_t(x), \quad s.t. \quad \mathcal{L}_\mathbb{K}(x) \leq 0, \tag{1}$$

where $\mathbb{K}$ is knowledge rules. Due to the structure of driving scenes, it is usually difficult to directly search $x$ in the data space, so we consider a generative model that creates $x$ with learnable parameters.

In this section, we first describe the tree-based generative model for learning the hierarchical representations of $x$, which is important and necessary for applying knowledge to achieve semantic controllability (Section 3.1). Then we explain the two types of knowledge to be integrated into the generative model together with the generation stage that uses explicit knowledge as constraints (Section 3.2).

### 3.1 Tree-structured Variational Auto-encoder (T-VAE)

VAE (Kingma & Welling, 2013) is a powerful model that combines auto-encoder and variational inference (Blei et al., 2017). It estimates a mapping between data point $x \in \mathcal{X}$ and latent code $z \in \mathcal{Z}$ to find the low-dimensional manifold of the data space. The objective function of training VAE is to maximize a lower bound of the likelihood of training data, which is the so-called Evidence Lower Bound (ELBO)

$$\text{ELBO} = \mathbb{E}_{q(z|x;\phi)}\left[\log p(x|z;\theta)\right] - \mathbb{KL}(q(z|x;\phi)||p(z)), \tag{2}$$

where $\mathbb{KL}$ is Kullback–Leibler (KL) divergence. $q(z|x;\phi)$ is the encoder with parameters $\phi$, and $p(x|z;\theta)$ is the decoder with parameters $\theta$. The prior distribution of the latent code $p(z)$ is usually a Gaussian distribution for simplification of KL divergence calculation.

#### 3.1.1 Tree structure design

One typical characteristic of driving scenes is that the data dimension varies with the number of objects. Thus, it is challenging to represent objects with a fixed number of parameters as in traditional models (Kingma & Welling, 2013). Graphs are commonly used to represent structured data (Liao et al., 2019) but are too complicated to describe the hierarchy and inefficient to generate. As a special case of graphs, trees naturally embed hierarchical information via recursive generation with depth-first-search traversal (Jin et al., 2018; Mo et al., 2020). This hierarchy is highly consistent with natural physical scenes and makes it easier to apply explicit knowledge, supported by previous works in cognition literature (Malcolm et al., 2016).

In this work, we propose a novel tree generative model that handles scenes with varying numbers of objects. Assume we have a stick with length $W$ and we recursively break it into segments $w^{(n,i)}$ with

$$W = w^{(1,1)} = w^{(2,1)} + w^{(2,2)} = \cdots = \sum_{i=1}^{K_n} w^{(n,i)}, \tag{3}$$

where $(n,i)$ means the $i$-th segment of the $n$-th break. $K_n$ is the total number of segments in the $n$-th break. The index starts from 1 thus index $(1,1)$ means the entire stick. The recursive function of breaking the stick

$$w^{(n+1,j)} = \alpha^{(n,i)} w^{(n,i)}, \quad w^{(n+1,j+1)} = (1 - \alpha^{(n,i)}) w^{(n,i)}, \tag{4}$$

where $\alpha^{(n,i)} \in [0,1]$ is the splitting ratio for segment $w^{(n,i)}$. Segment $w^{(n+1,j)}$ is the first segment of $w^{(n,i)}$ in the $(n+1)$-th break. Intuitively, this breaking process creates a tree structure where segments are nodes in the tree and the $i$-th break is corresponding to the $i$-th layer of the tree.

We extend the above division to 2-dimensional space as shown in the left of Figure. 1. To generate trees of driving scenes, we define three types of nodes, namely Quad (generates four child nodes), Object (describes one kind of object), and Empty (works as a placeholder). When there is more than one object in the region, the Quad node is used to divide the region and expand the tree to one more depth. Since the expansion always has four child nodes but not all nodes contain objects, the Empty node is used for filling the region with no object. If there is only one object in the region, the Object node is used to represent the property of this object and end the expansion of the tree. Different types of nodes can appear in the same layer and we follow Recursive Neural Networks (RvNN) (Socher et al., 2011) to build the tree structure recursively. Please refer to Appendix A.2 for a detailed example.

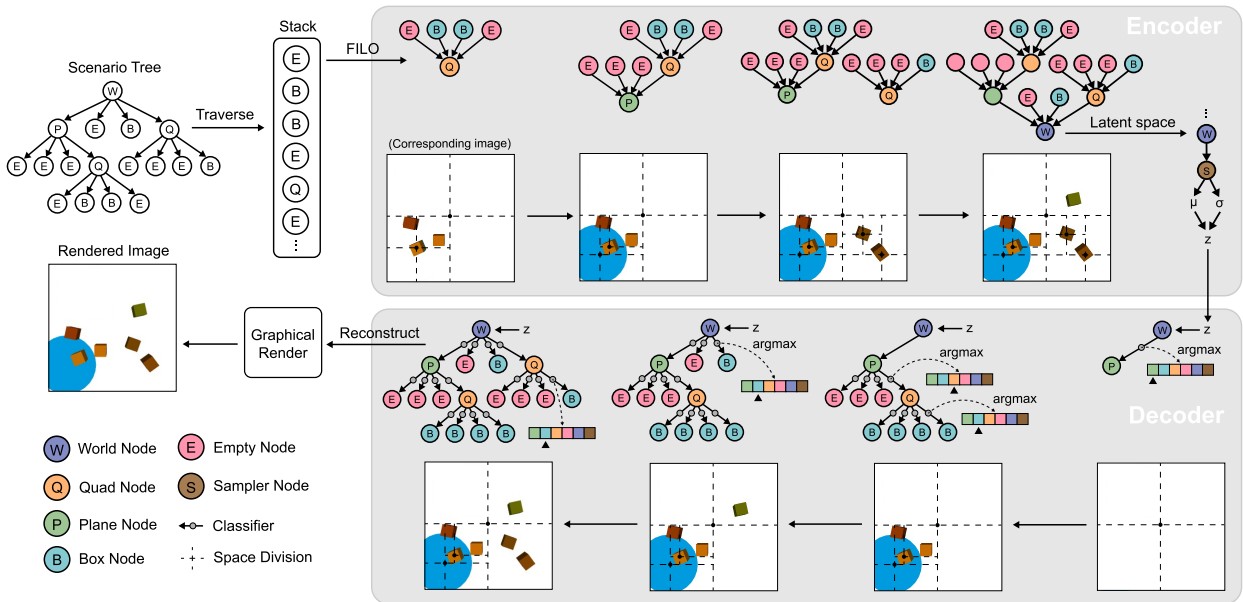

Figure 2: An example of the encoding and decoding processes of the proposed T-VAE model.

### 3.1.2 Model Implementation

We now introduce how to implement the encoding and decoding processes for a tree structure. Assuming there are $M$ types of objects (including Quad and Empty) in the scene, we use a set of encoders $\{E_m\}_{m=1}^{M}$ and decoders $\{D_m\}_{m=1}^{M}$ to construct each kind of nodes in the entire $N$ layer tree. $\{E_m\}_{m=1}^{M}$ create the *encoder tree* in a bottom-up manner and end at the latent variables $z$, while $\{D_m\}_{m=1}^{M}$ reconstruct the *decoder tree* in a top-down manner. The relationship between the $n$-th layer and the $(n+1)$-th layer in the *encoder tree* and the *decoder tree* are respectively:

$$
\begin{aligned}
f^{(n,i)} &= E_m([f^{(n+1,j)}, \cdots, f^{(n+1,j+3)}, g^{(n,i)}]; \phi_m), \\
[\hat{f}^{(n+1,j)}, \cdots, \hat{f}^{(n+1,j+3)}, \hat{g}^{(n,i)}] &= D_m(\hat{f}^{(n,i)}; \theta_m),
\end{aligned}
\tag{5}
$$

where $f^{(n,i)}$ is named as the feature vector that passes the messages through the tree structure. $g^{(n,i)}$ is named as the property vector of node $(n,i)$ that stories properties such as the color of the object generated by node $(n,i)$. In the bottom-up *encoder tree*, the selection of node type is accessible in the structured data, while in the top-down *decoder tree*, the selection does not have the reference. Therefore, a *Classifier* is required to determine the child node type $\hat{c}^{(n,i)}$:

$$
\hat{c}^{(n,i)} = \mathrm{Classifier}(f^{(n,i)}; \theta_c).
\tag{6}
$$

Between the encoders and decoders, the latent code $z$ is sampled according to parameters $[z_\mu, z_\sigma]$, which are estimated by a *Sampler* by the reparameterization trick (Blei et al., 2017):

$$
[z_\mu, z_\sigma] = \mathrm{Sampler}(f^{(1,1)}; \phi_s).
\tag{7}
$$

Finally, we can summarize all model parameters with $q(z|x; \phi)$ and $p(x|z; \theta)$, where $\phi = \{\phi_1, \cdots, \phi_m, \phi_s\}$ and $\theta = \{\theta_1, \cdots, \theta_m, \theta_c\}$.

### 3.1.3 Model Training

According to the implementation introduced above, the input scene $x$ to the *encoder tree* can be represented by the node type $c$ and property $g$ of all nodes.

$$
x = \{c, g\} = \{c^{(1,1)}, \cdots, c^{(N,K_N)}, \cdots, g^{(1,1)}, \cdots, g^{(N,K_N)}\},
\tag{8}
$$

Correspondingly, the output from the *decoder tree* is $\hat{x} = \{\hat{c}, \hat{g}\}$ Assume $c$ and $g$ are conditionally independent given $z$, we get the objective of T-VAE following the ELBO of VAE (2)

$$\max_{\phi,\theta} \quad \underbrace{\mathbb{E}_q \left[log\, p(c|z;\theta)\right]}_{-\mathcal{L}_C(\hat{c},c)} + \underbrace{\mathbb{E}_q \left[log\, p(g|z;\theta)\right]}_{-\mathcal{L}_R(\hat{g},g)} - \mathbb{KL} \left(\mathcal{N}(z_\mu, z_\sigma) \| \mathcal{N}(0, \mathbf{I})\right). \tag{9}$$

The first term $\mathcal{L}_C$ represents the cross-entropy loss (CE) of the *Classifier*

$$\mathcal{L}_C(\hat{c}, c) = \frac{1}{\sum_n^N K_n} \sum_{n=1}^{N} \sum_{i=1}^{K_n} \text{CE}(\hat{c}^{(n,i)}, c^{(n,i)}). \tag{10}$$

To make the *decoder tree* have the same structure as the *encoder tree*, we use Teacher Forcing (Williams & Zipser, 1989) during the training stage. However, in the generation stage, we select the node with the maximum probability as the child to expand the tree. The second term $\mathcal{L}_R$ uses mean square error (MSE) to approximate the log-likelihood of node properties from all decoders

$$\mathcal{L}_R(\hat{g}, g) = \sum_{m=1}^{M} \frac{1}{N_m} \sum_{n=1}^{N} \sum_{i=1}^{K_n} \mathbb{1} \left[ c^{(n,i)} = m \right] \|\hat{g}^{(n,i)} - g^{(n,i)}\|_2^2, \tag{11}$$

where $N_m$ is the times that node type $m$ appears in the tree and $\mathbb{1}[\cdot]$ is the indicator function. In (11), we normalize the MSE with $N_m$ instead of $\sum_n^N K_n$ to avoid the influence caused by imbalanced node type in the tree. To help understand the encoding and decoding process, we use a synthetic scene as an example to show the encoding and decoding processes in Figure 2, where the original data point is stored in the tree structure and the generated data is also a tree structure.

The advantage of this hierarchical structure is that the root node stores the global information, and other nodes only contain local information, making it easier for the model to capture the feature from multiple scales in the scene. In addition, this tree structure makes it possible to explicitly apply semantic knowledge in the generation stage, which will be explained in Section 3.2.

### 3.2 Knowledge-guided Generation

In the generation stage, we aim to create an adversarial scene $x$ to decrease $\mathcal{L}_t(x)$ by searching in the latent space of the *decoder tree* obtained in the previous training stage. Meanwhile, we use the knowledge $\mathbb{K}$, which represents traffic rules, to guide the search for a low knowledge loss $\mathcal{L}_\mathbb{K}$. We formulate this process as a constraint optimization problem in the latent space that uses the knowledge $\mathcal{L}_\mathbb{K} \leq 0$ as constraints to minimize $\mathcal{L}_t(x)$. The general idea is shown in Figure. 3(b).

---

**Algorithm 1:** Apply Knowledge

**Input:** $\mathbb{K}$, Decoder tree $\hat{x}$
**Output:** Modified decoder tree $\hat{x}'$
**Function** `ApplyK(`$\mathbb{K}$`, `$\hat{x}$`)`
**for** *each knowledge* $k^{(n)} \in \mathbb{K}$ **do**
  | $\hat{x}' \leftarrow$ modify $\hat{x}$ according to $k^{(n)}$
**end**
**if** *x has child nodes* **then**
  | **for** *all child nodes* $\hat{x}_i$ *of* $\hat{x}$ **do**
  |   | $\hat{x}_i' \leftarrow$ `ApplyK(`$\mathbb{K}$`, `$\hat{x}_i$`)`
  |   | Add node $\hat{x}_i'$ as a child to $\hat{x}'$
  | **end**
**end**
**return** $\hat{x}'$

---

#### 3.2.1 Knowledge representation

We first provide a formal definition to describe the knowledge that we use in the decoder tree. Suppose there is a function set $\mathcal{F}$, where the function $f(A) \in \mathcal{F}$ returns true or false for a given input node $A$ of a tree $x$. Then, we define the two types of propositional knowledge $\mathbb{K}$ for a particular victim model $t$ using the first-order logic (Smullyan, 1995) as follows.

**Definition 1 (Knowledge Set)** *The node-level knowledge $k_n$ is denoted as $f(A)$ for a function $f \in \mathcal{F}$, where $A$ is a single node. The edge-level knowledge $k_e$ is denoted as $f_1(A) \to \forall j\, f_2(B_j)$ for two functions $f_1, f_2 \in \mathcal{F}$, where we apply knowledge $f_2$ to all child nodes $B_j$ of $A$. Then, The knowledge set is constructed as $\mathbb{K} = \{k_n^{(1)}, \cdots, k_e^{(1)}, \cdots\}$.*

In the tree context, $k_n$ describes the properties of a single node, and $k_e$ describes the relationship between the parent node and its children. Specifically, in order to satisfy $f(A)$ in $k_n$, we locate node $A$ in the tree $x$

---

**Algorithm 2:** SAG Framework

---

**Input:** Dataset $\mathcal{D}$, Task loss $\mathcal{L}_t(x)$, budget $B$,  
       Knowledge set $\mathbb{K}$  
**Output:** Generated scene $\hat{x}$  
**Stage 1:** `Train T-VAE`  
    Initialize model parameters $\{\theta, \phi\}$  
    **for** $x$ *in* $\mathcal{D}$ **do**  
        Encode $z \leftarrow q(z|x; \phi)$  
        Decode $\hat{x} \leftarrow p(x|z; \theta)$  
        Update parameters $\{\theta, \phi\}$ by maximizing  
        ELBO (9)  
    **end**  
Store the learned decoder $p(x|z; \theta)$

**Stage 2:** `Knowledge-guided Generation`  
    Initialize latent code $z \sim \mathcal{N}(0, \mathbf{I})$  
    **while** $B$ *is not used up* **do**  
        **if** $\mathcal{L}_t(x)$ *is differentiable* **then**  
           $z \leftarrow z - \eta \nabla \mathcal{L}_t(p(x|z; \theta))$  
        **else**  
           $z \leftarrow$ Black-box Optimization  
        **end**  
        $\hat{x}' \leftarrow$ `ApplyK`$(\mathbb{K}, \hat{x})$, $\hat{x} \leftarrow p(x|z; \theta)$  
        $z \leftarrow \mathbf{prox}_{\mathcal{L}_{\mathbb{K}}}(z, \hat{x}, \hat{x}')$ with (13)  
    **end**  
Decode the scene $\hat{x} = p(x|z; \theta)$

---

and change the property vector from $g$ to $g'$. Similarly, in order to satisfy $f_1(A) \to f_2(B_j)$, after traversing $x$ to find node $A$ that satisfy $f_1$ in $k_e$, we change the type vector from $c$ to $c'$ or the property vector from $g$ to $g'$ for all $A$'s children so that $f_2(B_j)$ holds true. The reference vectors $c'$ and $g'$ are pre-defined by the knowledge set $\mathbb{K}$. We summarized the process of applying knowledge in **Algorithm 1**.

One running example is that the explicit knowledge described as *"if one node represents blue, its child nodes should represent red"* is implemented by the following operations. Starting from the root, we find all nodes whose colors are blue and collect the property vectors $g$ of its child nodes; then we change $g$ to $g'$, representing the red color. This process is illustrated in Figure 3(a).

### 3.2.2 Adversarial Generation

To minimize the adversarial loss and satisfy the constraints of knowledge, we combine them to the new objective $\mathcal{L}(x) = \mathcal{L}_t(x) + \mathcal{L}_{\mathbb{K}}$, where the second term represents the mismatch between the original decoder tree $\hat{x}$ and the modified tree $\hat{x}'$ (shown in Figure 3(a))

$$\mathcal{L}_{\mathbb{K}}(\hat{x}, \hat{x}') = \mathrm{MSE}(\hat{g}_i, \hat{g}_i') + \mathrm{CE}(\hat{c}_i, \hat{c}_i'), \quad \forall \hat{x}_i \neq \hat{x}_i'. \tag{12}$$

Usually, $\mathcal{L}_t(x)$ requires large computations, while $\mathcal{L}_{\mathbb{K}}$ is efficient to evaluate since it only involves the inference of $p(x|z; \theta)$. Therefore, we resort to Proximal algorithms (Parikh & Boyd, 2014), which alternatively optimize $\mathcal{L}_t(x)$ and $\mathcal{L}_{\mathbb{K}}$. In our setting, explicit knowledge is regarded as the trusted region to guide the optimization of $\mathcal{L}_t(x)$. The knowledge loss and adversarial objective are alternatively optimized under the proximal optimization framework as shown in Figure 3(b). In the step of optimizing $\mathcal{L}_t(x)$ (pink

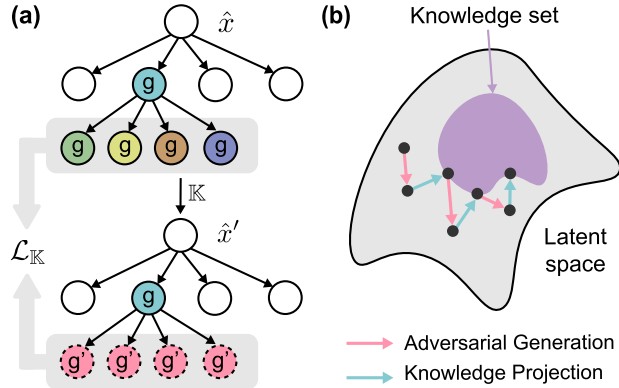

Figure 3: **(a)** The knowledge integration example described in Section 3.2: the child nodes of the blue color node should be red. **(b)** Illustration of the knowledge-guided generation process by proximal optimization.

arrow), we can either use gradient descent for a differentiable $\mathcal{L}_t(x)$ or change to black-box optimization methods (Audet & Hare, 2017) when $\mathcal{L}_t(x)$ is non-differentiable. Then, in the step of the optimizing $\mathcal{L}_{\mathbb{K}}$ (blue arrow), we use **Algorithm 1** to get the modified *decoder tree* $\hat{x}'$ and use the following proximal operator

$$z' = \mathbf{prox}_{\mathcal{L}_{\mathbb{K}}}(z, \hat{x}, \hat{x}') = \arg \min_{z'} \left( \mathcal{L}_{\mathbb{K}}(\hat{x}, \hat{x}') + \frac{1}{2} \|z - z'\|_2^2 \right) \tag{13}$$

to project the latent code $z$ to $z'$ so that $p(x|z'; \theta)$ satisfies the knowledge rules. The second term in (13) is a regularize to make the projected point also close to the original point. The equation (13) can be solved by gradient descent since the decoder $p(x|z; \theta)$ of T-VAE is differentiable. In summary, The entire training and generation stages are shown in **Algorithm 2**.

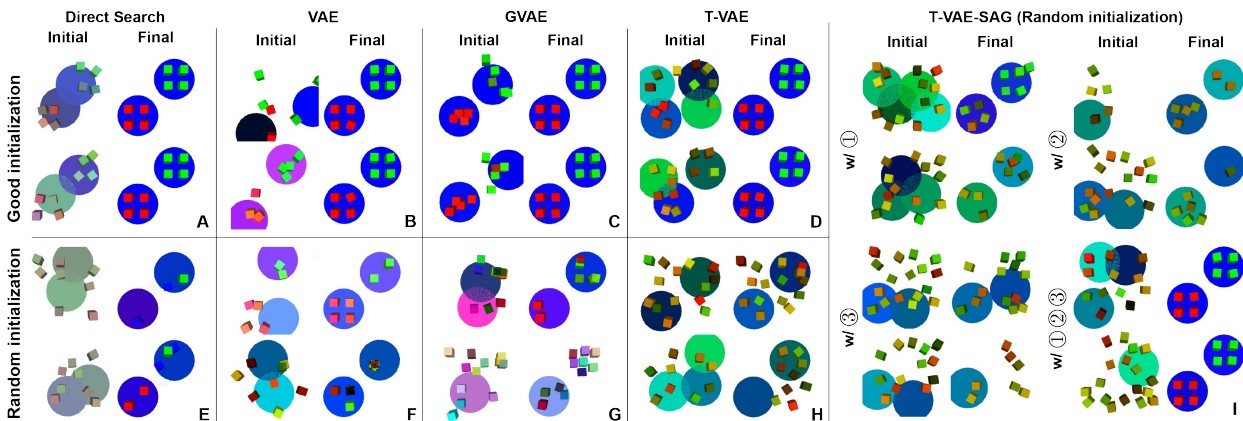

Figure 4: Results of synthetic scene reconstruction experiment from 5 methods with random and good initialization. **I** shows the results of T-VAE using SAG. With the combination of knowledge ①②③, we can almost reach the optimal solution even from a random initialization, while baseline methods can realize the target *only* when starting from the good initialization.

## 4 Experiments

In the experiment, we first design a synthetic scene to illustrate the controllability and explainability of the proposed framework. The synthetic physical scene provides a simplified setting to unveil the essence of the knowledge-guided generation. After that, we evaluate the performance of SAG on realistic driving scenes represented by point clouds. Based on SAG, we propose a new adversarial attack method, *Scene Attack*, against multiple point cloud segmentation methods.

### 4.1 Synthetic Scene Reconstruction

#### 4.1.1 Task description

In this task, we aim to reconstruct a scene to match a given image. The objective is the reconstruction error $\mathcal{L}_t(x) = \|S - \mathcal{R}(x)\|_2$, where $\mathcal{R}$ is a differentiable image renderer (Kato et al., 2018) and $S$ is the image of target scene. Under this succinct setting, it is possible to analyze and compare the contribution of explicit knowledge integration since we can access the optimal solution, which usually cannot be obtained in an adversarial attack. According to the understanding of the target scene, we define three knowledge rules: ① The scene has at most two plates; ② The boxes that belong to the same plate should have the same color; ③ The boxes belong to the same plate should have distance smaller than a threshold $\gamma$.

#### 4.1.2 Experiment settings

We synthesize the dataset by randomly generating 10,000 samples with a varying number of boxes and plates. We compare our method with the following baselines: Direct Search (**DS**) directly optimizes the positions and colors of boxes and plates in the data space. Direct Search with constraints (**DS-C**) modifies DS by adding knowledge constraints ①② to the objective function. **VAE** (Kingma & Welling, 2013) is a well-known generative model that supports the latent space searching for sample generation. **VAE-WR** (Tripp et al., 2020) simultaneously updates the shape of latent space during the searching process. **SPIRAL** (Ganin et al., 2018) generates one object at one time to create the scene in an autoregressive manner. **L2C** (Ding et al., 2020) uses autoregressive structure to generate objects in the scene. **Grammar-VAE (GVAE)** (Kusner et al., 2017) uses pre-defined rules (shown in Appendix A.4) to generate the structural scene. **T-VAE** only uses the tree structure to build the model; the searching is done in the latent space without any knowledge integration.

Among these methods, DS, DS-C, VAE, and VAE-WR need to access the number of boxes and plates in the target image (e.g., two plates and eight boxes) to fix the dimension of the input feature. To get good initial

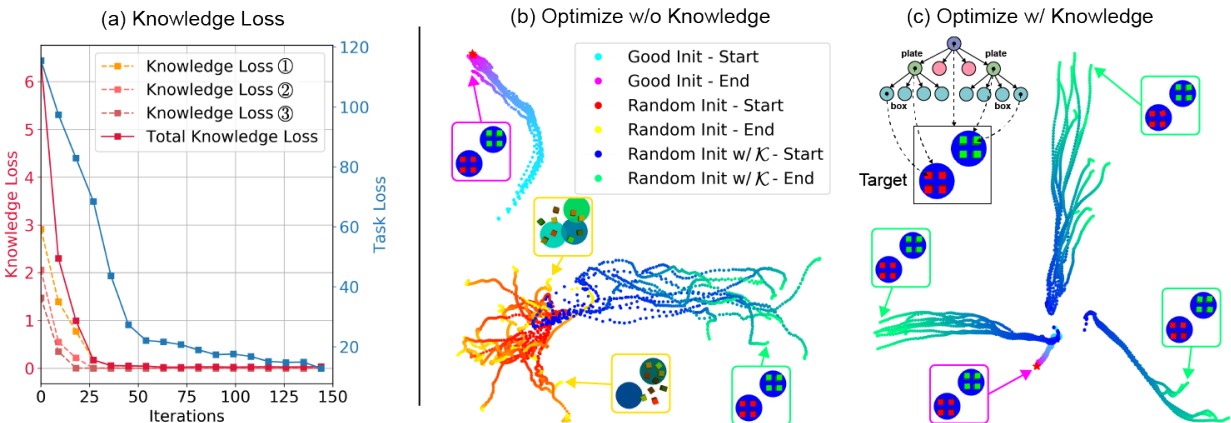

Figure 5: **(a)** Knowledge loss of integrating semantic rules ①②③ separately. **(b)** The influence of knowledge on the trajectories with the same initialization. **(c)** After applying explicit knowledge, optimization trajectories are diverse when starting from different initialization.

points for DS and DS-C, we add a small perturbation to the positions and colors of all objects in the target scene. Similarly, we add the perturbation to the optimal latent code for other methods, which is obtained by passing the target scene to the encoder to get good initialization.

### 4.1.3 Evaluation results

We show the generated samples from five representative methods in Figure 4 and show the final errors of all methods in Table. 1. With good initialization, all models find a similar scene to the target one, while with random initialization, all models are trapped in local minimums. However, obtaining good initialization is not practical in most real-world applications, indicating that this task is non-trivial and all models without knowledge cannot solve it. After integrating the knowledge into the T-VAE model, we obtain **I** of Figure 4. We can see that all three knowledge have positive guidance for the optimization, e.g., the boxes concentrate on the centers of plates with knowledge ③ When combining the three rules of knowledge, even from a random initialization, our T-VAE can finally find the target scene, leading to a small error in Table. 1. We also want to mention that it is also possible to apply simple

Table 1: Reconstruction Error

| Method | Initialization | |
|---|---|---|
| | Random | Good |
| DS | 86.0±9.4 | **7.9±1.2** |
| DS-C | 90.6±13.1 | 8.1±1.4 |
| VAE | 110.4±10.6 | 13.4±6.1 |
| VAE-WR | 105.9±24.6 | 13.2±8.4 |
| SPIRAL | 95.2±21.9 | 23.6±5.5 |
| L2C | 115.4±13.8 | 14.1±7.1 |
| GVAE | 123.7±9.5 | 19.7±10.2 |
| T-VAE | 135.1±16.9 | 14.1±2.5 |
| T-VAE-SAG | **14.5±1.3** | 11.8±2.1 |

knowledge to GVAE during the generation. However, the advantages of our method are that (1) we can integrate any constraints as long as they can be represented by Definition 1. In contrast, GVAE can only apply hard constraints to objects with co-occurrences.

### 4.1.4 Analysis of knowledge and controllability

To analyze the contribution of each knowledge, we plot the knowledge losses of ①②③ in Figure 5(a) together with the adversarial loss. All knowledge losses decrease quickly at the beginning and guide the search in the latent space. Next, we made ablation studies to explore why knowledge helps the generation. In Figure 5(b), we compare the optimization trajectories of T-VAE (red→yellow) and T-VAE-SAG (blue→green) with the same initialization. For T-VAE, the generated samples are diverse but totally different from the target scene, while for T-VAE-SAG, the trajectories go in another direction and the generated samples are good. However, note that although the knowledge helps us find good scenes, it does not reach the same point in the latent space with the trajectories from good initialization (cyan→purple).

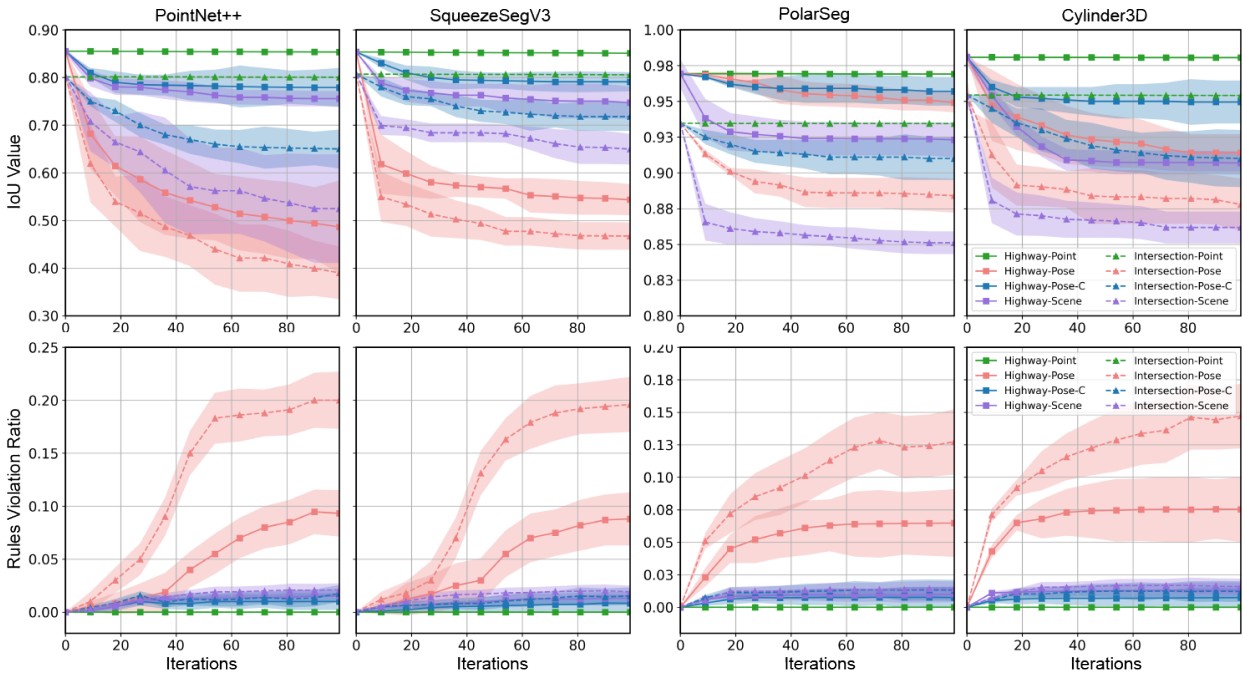

Figure 6: **Top:** The IoU values during the attack process of four victim models on two backgrounds. **Bottom:** The ratio of rules violation of all methods, on two backgrounds. Although *Pose Attack* outperforms our *Scene Attack* in terms of the IoU value in some cases, it also has a large ratio of rules violation, which means the scenes generated by *Pose Attack* are not realistic as shown in Figure 7.

This result can be explained by the entanglement of the latent space (Locatello et al., 2019), which makes multiple variables control the same property. To further study this phenomenon, we plot Figure 5(c), where we use 3 different initialization for T-VAE-SAG. The result shows that all three cases find the target scene but with totally different trajectories, which supports our conjecture. *In summary, we believe the contribution of knowledge can be attributed to the entanglement of the latent space, which makes the searching easily escape the local minimum and find the nearest optimal points.*

## 4.2 Adversarial Driving Scenes Generation

### 4.2.1 Task description

In this task, we aim to generate *realistic* driving scenes against segmentation algorithms as well as satisfy specific semantic knowledge rules. The adversarial scenes are defined as scenes that reduce the performance of victim models. To generate adversarial LiDAR scenes containing various fore-/background rather than the point cloud of a single 3D object as existing studies (Lang et al., 2020; Sun et al., 2020b), a couple of challenges should be considered: First, LiDAR scenes with millions of points are hard to be directly operated; Second, generated scenes need to be realistic and follow traffic rules. Since there are no existing methods to compare with directly, we compare three methods: (1) *Point Attack*: a point-wise attack baseline (Xiang et al., 2019) that adds small disturbance to points; (2) *Pose Attack*: a scene generation method developed by us that searches pose of vehicles in the scene; (3) *Scene Attack*: a semantically controllable generative method based on our T-VAE and SAG.

We explore the attack effectiveness against different models of these methods and their transferability. For *Pose Attack* and *Scene Attack*, we implement an efficient LiDAR model $\mathcal{R}(x, B)$ (Möller & Trumbore, 1997) (refer to Appendix A.1 for details) to convert the generated scene $x$ to a point cloud scene with a background $B$. The task objective $\min \mathcal{L}_t(x) = \max \mathcal{L}_P(\mathcal{R}(x, B))$ is defined by maximizing the loss function $\mathcal{L}_P$ of segmentation algorithms $P$. We design three explicit knowledge rules: ① roads follow a given layout (location, width, and length); ② vehicles on the lane follow the direction of the lane; ③ vehicles should gather together

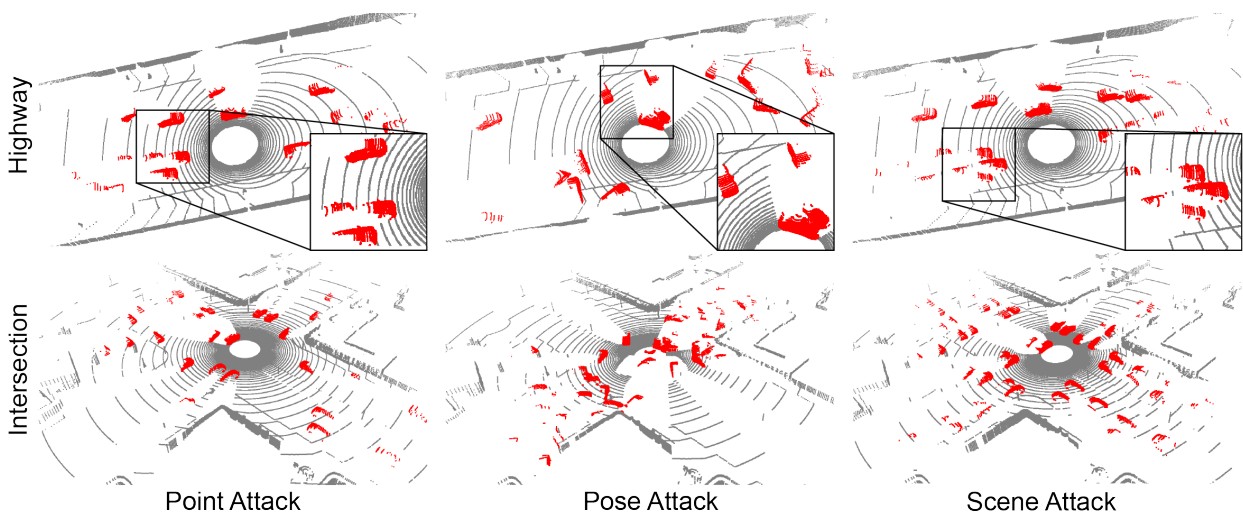

Figure 7: Scenarios from three adversarial generation methods for PointNet++ model in *Highway* and *Intersection* backgrounds. Red points represent vehicles. Scenes generated by *Scene Attack* are complicated and follow basic traffic rules, while scenes generated by *Pose Attack* violate physical laws and traffic rules.

but keep a certain distance. ①② ensure generated vehicles follow the layout of the background $B$ and ③ makes the scene contain more vehicles.

### 4.2.2 Experiment settings

We select four point cloud segmentation algorithms, PointNet++ (Qi et al., 2017), PolarSeg (Zhang et al., 2020), SqueezeSegV3 (Xu et al., 2020), Cylinder3D (Zhou et al., 2020) as our victim models, all of which are pre-trained on the Semantic Kitti dataset (Behley et al., 2019). We use two backgrounds $B$ (Highway and Intersection) collected from the CARLA simulator (Dosovitskiy et al., 2017) as scene templates. Since it is usually unable to access the parameters of segmentation algorithms, we focus on the black-box attack in this task. The *Point Attack* optimizes $\mathcal{L}_t(x)$ with SimBA (Guo et al., 2019), while *Pose Attack* and *Scene Attack* optimizes $\mathcal{L}_t(x)$ with Bayesian Optimization (BO) (Pelikan et al., 1999). For the training of T-VAE, we build a dataset by extracting the pose information of vehicles together with road and lane information from the Argoverse dataset (Chang et al., 2019).

Table 2: Transferability of Adversarial Scenes (Point Attack IoU / Scene Attack IoU). *Scene Attack* has lower IoU for all evaluation pairs, which demonstrates its better adversarial transferability.

| Source \ Target | PointNet++ | SqueezeSegV3 | PolarSeg | Cylinder3D |
|---|---|---|---|---|
| PointNet++[*] | - / - | 0.916 / **0.768** | 0.936 / **0.854** | 0.955 / **0.918** |
| SqueezeSegV3 | 0.954 / **0.606** | - / - | 0.932 / **0.855** | 0.956 / **0.892** |
| PolarSeg | 0.952 / **0.528** | 0.904 / **0.753** | - / - | 0.953 / **0.908** |
| Cylinder3D | 0.951 / **0.507** | 0.903 / **0.688** | 0.934 / **0.877** | - / - |

[*] The IoU for *Point Attack* is obtained after 20,000 iterations.

### 4.2.3 Result Analysis

**Effective of Adversarial Attack.** We use Intersection over Union (IoU) for the vehicle as the metric to indicate the performance of segmentation algorithms. We show the results as well as the ratio of rules violation (ratio of objects that violate knowledge ①②) during the attack in Figure 6. Generally, it is harder to find adversarial scenes in the highway background than in the intersection background since the latter has much more vehicles. Within 100 iterations, the *Point Attack* method nearly has no influence on the

Table 3: Detection Results (IoU) with Adversarial Training

| Method | Pose | Pose w/ AT | Pose-C | Pose-C w/ AT | SAG | SAG w/ AT |
|---|---|---|---|---|---|---|
| PointNet++ | 0.3902 | 0.1315 | 0.6500 | 0.6429 | 0.5248 | **0.7657** |
| SqueezeSeg | 0.4672 | 0.0974 | 0.7180 | 0.7378 | 0.6496 | **0.8239** |
| PolarSeg | 0.8840 | 0.0106 | 0.9112 | 0.9073 | 0.8511 | **0.8923** |
| Cylinder3D | 0.8779 | 0.0806 | 0.9032 | 0.9004 | 0.8618 | **0.9120** |

performance since it operates in very high dimensions. In contrast, *Pose Attack* and *Scene Attack* efficiently reduce the IoU value. Although *Pose Attack* achieves comparable results to our method, scenes generated by it (shown in Figure 6) are unrealistic due to the overlaps between vehicles; therefore, the ratio of rules violation is high. In contrast, scenes generated by our method only modify the vehicles within the traffic constraints. More generated scenes can be found in Appendix A.5.

**Transferability Analysis.** In Table 2, we show the transferability of *Point Attack* and *Scene Attack*. Transferability means using generated samples from the *Source* model to attack other *Target* models, which is crucial for evaluating adversarial attack algorithms. Although *Point Attack* dramatically reduce the performance of all four victims, the generated scenes have weak transferability since it cannot attack other victim models. However, scenes generated by *Scene Attack* successfully attack all models, even those that are not used during the training, showing strong adversarial transferability.

**Adversarial Training.** In Table 3, we further explore the performance of adversarial training using scenes generated from different methods. We find that training algorithms with scenes from Pose and Pose-C even reduces the performance. This is because generated scenes are not in the same distribution as the original training data since they do not satisfy physical laws and traffic rules. In contrast, training algorithms with scenes from our SAG improves the robustness of all algorithms against adversarial attacks, which shows one promising usage of our scene generation method.

## 5 Conclusion

In this paper, we explore semantically adversarial generation tasks with explicit knowledge integration. Inspired by the categorization of knowledge for the driving scene description, we design a tree-structured generative model to represent structured data. We show that the two types of knowledge can be explicitly injected into the tree structure to guide and restrict the generation process efficiently and effectively. After considering explicit semantic knowledge, we verify that the generated data contain dramatically fewer semantic constraint violations. Meanwhile, the generated data still maintain the diversity property and follow the original data distribution. Although we focus on the scene generation application, the SAG framework can be extended to other structured data generation tasks, such as chemical molecules and programming languages, showing the hierarchical properties. One limitation of this work is that we assume the knowledge is helpful or at least harmless as they are summarized and provided by domain experts. However, the correctness of knowledge needs careful examination in the future.

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

# A    Appendix

The appendices are organized as follows:

- In **Appendix A.1**, we provide the details of our LiDAR model implementation used in the traffic scene generation experiment.

- In **Appendix A.2**, we provide the details of the structure of our proposed T-VAE model, including the definitions of all encoder-decoder pairs and a generation example.

- In **Appendix A.3**, we show the details of the definition of knowledge in both experiments.

- In **Appendix A.4**, we describe the three baselines used in the *Synthetic Scene Reconstruction* experiment.

- In **Appendix A.5**, we provide more experiment results.

- In **Appendix A.6**, we describe four point cloud segmentation victim models used in the *LiDAR Scene Generation* experiment.

## A.1    LiDAR Model Implementation

The LiDAR model is implemented by Moller-Trumbore algorithm Möller & Trumbore (1997) with the PyTorch Paszke et al. (2019) package for high-efficiency computation. We assume there is a plane $\triangle V_0 V_1 V_2$ in the 3D space constructed by points $V_0$, $V_1$, $V_2$. A ray $R(t)$ with origin $O$ and normalized direction $D$ (we make $\|D\|_2 = 1$ for simplification) is represented as $R(t) = O + tD$, where $t$ is the distance between $O$ and the endpoint $D$ of the ray. If $D$ is the intersection between the ray and the plane $\triangle V_0 V_1 V_2$, we can represent the ray with barycentric coordinate:

$$T(u,v) = (1 - u - v)V_0 + uV_1 + vV_2 = O + tD = R(t) \tag{14}$$

where $u$ and $v$ are weights. To simplify the equations, we define three new notations:

$$\begin{cases} E_1 = V_1 - V_0 \\ E_2 = V_2 - V_0 \\ T = O - V_0 \end{cases} \tag{15}$$

Then, we can solve the distance $t$ in (14) by:

$$\begin{bmatrix} t \\ u \\ v \end{bmatrix} = \frac{1}{|-D, E_1, E_2|} \begin{bmatrix} |-T, E_1, E_2| \\ |-D, T, E_2| \\ |-D, E_1, T| \end{bmatrix} = \frac{1}{(D \times E_2) \cdot E_1} \begin{bmatrix} (T \times E_1) \cdot E_2 \\ (D \times E_2) \cdot T \\ (T \times E_1) \cdot D \end{bmatrix} \tag{16}$$

Since we know the ray direction $D$, we can get the 3D coordinate of the intersection with this distance $t$. During the implementation, we reuse $D \times E_2$ and $T \times E_1$ to speed up the computation. To make sure the intersecting point $T(u,v)$ is inside the triangle $\triangle V_0 V_1 V_2$, we need to have:

$$u, v, (1 - u - v) \in [0, 1] \tag{17}$$

If these three conditions are not fulfilled, the intersection point will be removed.

To calculate the point cloud generated by a LiDAR, we first convert vehicle mesh models to triangles $\mathcal{F}$ with Delaunay triangulation Lee & Schachter (1980). Then, we create the array of LiDAR rays with width $W$ and height $H$ for 360° view. Finally, we use (16) to calculate the intersection point between all triangles $\mathcal{F}$ and LiDAR rays $R_{i,j}(t)$ in parallel to get the final range map $F^{H \times W}$. The background point cloud will also be converted to range map $B^{H \times W}$ by:

$$\begin{cases} \theta = \arctan \dfrac{z}{\sqrt{x^2 + y^2}} \\ \phi = \arctan \dfrac{x}{y} \\ t = \sqrt{x^2 + y^2 + z^2} \end{cases} \tag{18}$$

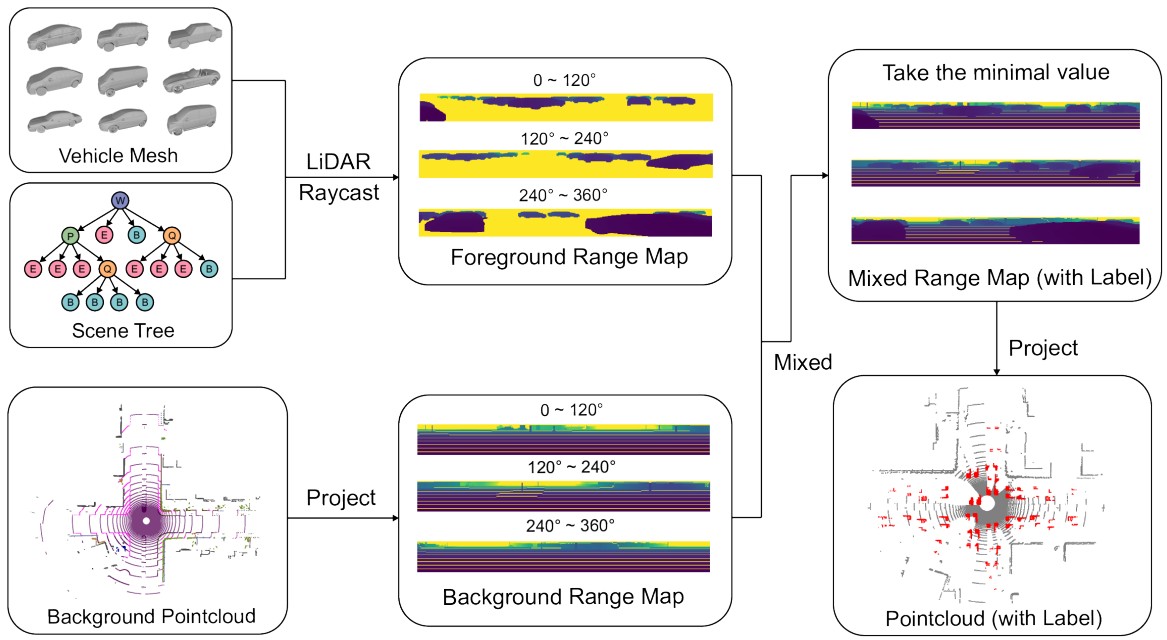

Figure 8: The pipeline of LiDAR scene generation with our developed model.

where $(x, y, z)$ is the coordinate of one point in point cloud, $\theta$ is used to calculate the index of the row, and $\phi$ is used to calculate the index of the column. Then we mix $F^{H \times W}$ and $B^{H \times W}$ by taking the minimal value for each element:

$$M_{i,j} = \min\{F_{i,j}, B_{i,j}\}, \quad \forall i \in W, \ j \in H \tag{19}$$

where $M_{i,j}$ represent the $(i, j)$-entry of the mixed range map scene $M$. Then, we convert the range map to the final output point cloud scene $S$ with:

$$\begin{cases} z = t \times \sin\theta \\ x = t \times \cos\theta\cos\phi \\ y = t \times \cos\theta\sin\phi \end{cases} \tag{20}$$

The entire pipeline of the above process is summarized in Figure 8. The parameters we used for LiDAR follow the configuration of the Semantic Kitti dataset, where the channel $H = 64$, the horizontal resolution $W = 2048$, the upper angle is $2°$, and the lower angle is $-25°$. The gap between reality and simulation can be reduced by realistic simulation and sensor models Manivasagam et al. (2020), but this will not be explored in this paper.

## A.2  Detailed T-VAE Model Structure

Our T-VAE model consists of several encoders and decoders that are related to the definition of the scene. In this paper, we explored two experiments with two scenes: a synthetic box placement image scene and a traffic point cloud scene. In Figure 9, we show the details of the modules for two scenes. The encoding process converts the tree into a stack and then encodes the information using the node type defined in the tree. The decoding process expands the tree with the predicted node type from the *Classifier*.

For the synthetic box placement image scene, there are 5 *Encoder-Decoder* pairs, a *Sampler*, and a *Classifier*. W node determines the global information such as the location and orientation of the entire scene. P node spawns a plate object in the scene with positions and colors determined by the property vector. Both Q and B nodes spawn a box object in the scene with positions and colors determined by the property vector. E node serves as a stop signal to end the expansion of a branch, therefore, will not spawn anything in the scene and does not have model parameters.

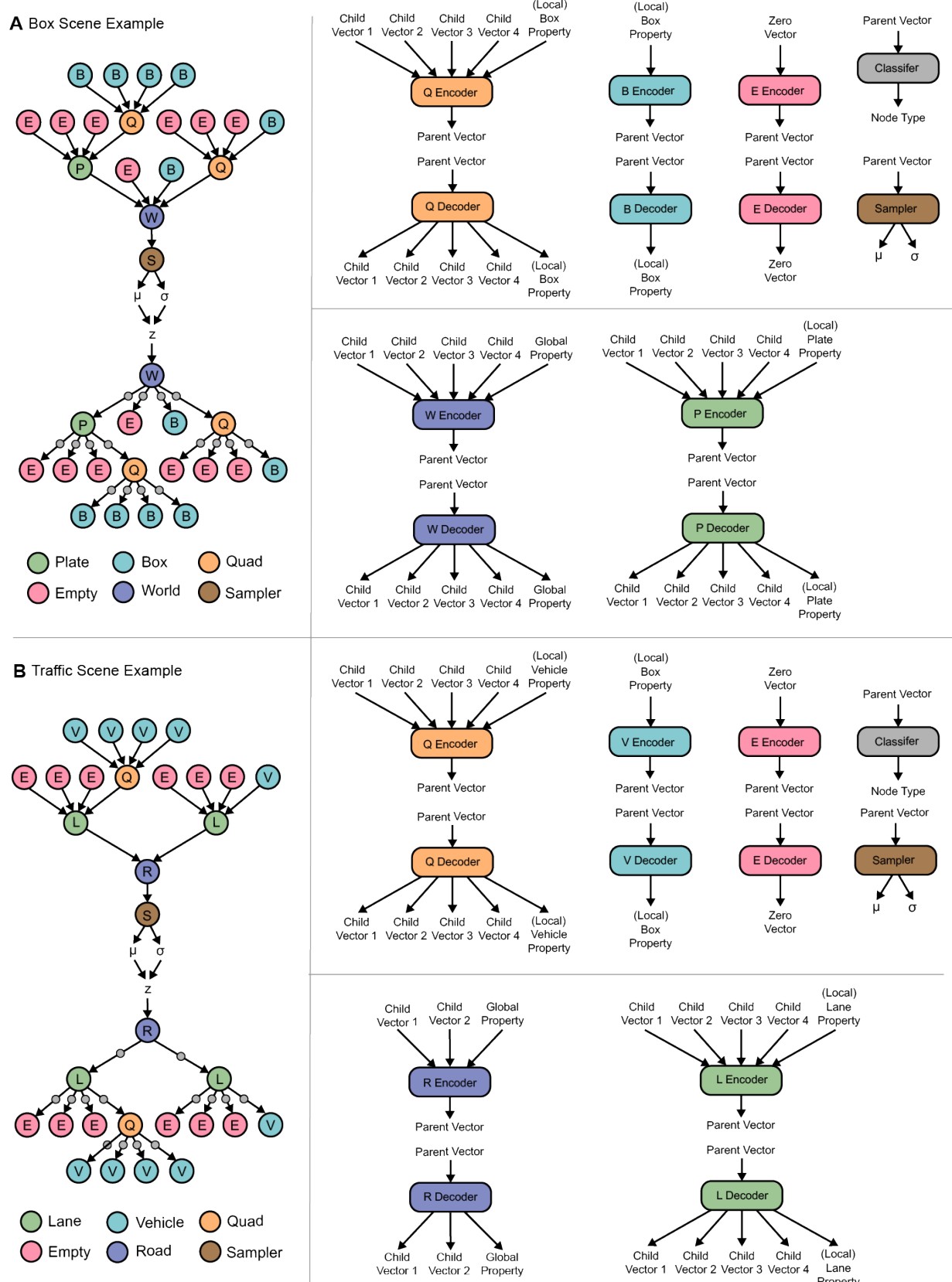

Figure 9: The definition of each module in our proposed T-VAE. **A**: There are 5 kinds of nodes in Synthetic Scene Reconstruction experiment. Therefore, we have 5 encoders and 5 decoders in total, plus a *Classifier* and a *Sampler*. **B**: There are 5 kinds of nodes in the LiDAR scene experiment. Therefore, we have 5 encoders and 5 decoders in total, plus a *Classifier* and a *Sampler*.

Table 4: Hyper-parameters of the Synthetic Scene Reconstruction Experiment

| Parameters | Value | Description |
|---|---|---|
| $lr$ | 0.001 | Learning rate of T-VAE training |
| $E$ | 1000 | Maximum training epoch |
| $B$ | 128 | Batch size during training |
| $\eta$ | 0.1 | Learning rate in stage 2. |
| $T$ | 100 | Maximum searching iteration |
| $d_z$ | 64 | Dimension of latent code $z$ |
| $d_f$ | 128 | Dimension of feature vector $f$ |
| $d_g$ | 6 | Dimension of property vector $g$ |
| $\gamma$ | 2 | The threshold used in knowledge ③ |
| $N_l$ | 10 | Normalization factor of location |

Table 5: Hyper-parameters of the LiDAR Scene Generation Experiment

| Parameters | Value | Description |
|---|---|---|
| $lr$ | 0.001 | Learning rate of T-VAE training |
| $\epsilon$ | 0.01 | Max value for point-wise disturbance |
| $E$ | 1000 | Maximum training epoch |
| $B$ | 128 | Batch size during training |
| $T$ | 100 | Maximum searching iteration |
| $d_z$ | 32 | dimension of latent code $z$ |
| $d_f$ | 64 | dimension of feature vector $f$ |
| $d_g$ | 3 | dimension of property vector $g$ |
| $N_l$ | 40 | Normalization factor of location |
| $(w, h)$ | (1.5, 3) | The thresholds used in knowledge ③ |

The traffic point cloud scene has similar definitions. R node contains the information about the road and only has two children. L node determines the lane information such as the width and direction. Both Q and V nodes spawn a vehicle in the scene with positions and orientations determined by the property vector.

### A.3 Knowledge Definition

For each experiment in this paper, we design three knowledge rules. We explain the details of the implementation of these rules.

In the Synthetic Scene Reconstruction Experiment, we calculate $\mathcal{L}_Y(x, Y_t(x))$ with the following implementations:

① *The scene has at most two plates*, which can be implemented by *W node has at most two P children nodes.* We traverse the entire generated tree $x$ to find $W$ node, then we collect the children nodes of $W$ and count the number of $P$ nodes. If the number is larger than 2, we calculate the cross-entropy loss between the node type and $E$ node label.

② *The colors of the boxes that belong to the same plate should be the same*, which can be implemented by *The color of P node's children nodes should be the same.* We traverse the entire generated tree $x$ to find all $P$ nodes, then we collect the colors of the children of $P$. The average color $\bar{c}$ is calculated for each $P$ node and $\bar{c}$ is used as a label to calculate the MSE for all children nodes of the corresponding $P$ node.

③ *The distance between the boxes that belong to the same plate should be smaller than a threshold $\gamma$* which can be implemented by *The distance between P node's children nodes should be smaller than a threshold $\gamma$.* We traverse the entire generated tree $x$ to find $P$ node, then we collect the absolute position of all its children nodes and calculate the MSE between this position and the position of $P$.

In the LiDAR Scene Generation Experiment, we calculate $\mathcal{L}_Y(x, Y_t(x))$ with the following implementations:

① *Roads follow a given layout (location, width, and length)*, which can be implemented by *R node follows a given layout (location, width, and length)*. We traverse the entire generated tree $x$ to find $R$ node, then calculate the MSE between the property vector of $R$ and the given layout.

② *Vehicles on the lane follow the direction of the lane*, which can be implemented by *L node follows pre-defined directions*. We traverse the entire generated tree $x$ to find $L$ node, then calculate the MSE between the property vector of $R$ and the given layout.

③ *Vehicles should gather together but keep a certain distance*, which can implemented by *Q node has at least two Q nodes as its children until the absolute width and height of the current block is smaller than thresholds $w$ and $h$*. We traverse the entire generated tree $x$ to find $Q$ node, then collect the type of its children nodes. When the collected $Q$ node type is less than 2, we calculate the cross-entropy loss between two collected node types and $Q$ node label. When the width and the height of current block are smaller than $w$ and $h$, we stop applying this rule.

After calculating all errors in $\mathcal{L}_Y(x, Y_t(x))$, we can directly use back-prorogation to calculate the gradient of latent code $z$ and update it with the gradient descent method.

### A.4    Baselines in Synthetic Scene Reconstruction

### A.4.1    Direct Search

The dimension of the physical space is $6 \times (2 + 8) = 60$, where 6 is the property dimension including position, orientation, and colors, 2 is the number of plates, and 8 is the number of boxes. We use gradient descent with the learning rate $\eta$ to directly search in the physical space. Since there is no constraint to avoid overlaps between boxes, the generated scenes could be unrealistic.

### A.4.2    Variational Auto-encoder (VAE)

The input dimension of the encoder is the same as the searching space of *Direct Search*. The VAE model has an encoder and a decoder, both of which have a fixed number of model parameters. There are 4 hidden layers in the encoder and each layer has 128 neurons. The decoder also has 4 hidden layers with 128 neurons. For the output of color, we add a $Sigmoid(\cdot)$ function to normalize it to the range $[0, 1]$. The location is normalized by $N_l$ before it is taken into the encoder.

### A.4.3    Grammar VAE (GVAE)

GVAE Kusner et al. (2017) requires the input data to be described with a set of pre-defined grammars. According to the task of synthetic scene reconstruction experiment, we design 9 rules,

$$W \to P, \ W \to B, \ P \to P|E, \ P \to P|B, \ P \to B,$$
$$P \to E, \ B \to B, \ B \to E, \ E \to E \tag{21}$$

and they are represented in a one-hot vector in the dataset. The original GVAE is designed only for rule-based discrete data generation (e.g. molecules), thus we modify the structure to add continuous attribute representation. The encoder consists of 3 1-dimensional Convolution layers with kernel size $3 \times 3$. The numbers of channels for the Convolution layers are $[32, 64, 128]$. The decoder is an LSTM model with 128 neurons, therefore, the decoding process is sequential. During the training stage, the maximal length of rules is fixed to 20 with $E \to E$ as padding, and the decoder will output 20 rules. The cross-entropy loss is used between the input rules and decoded rules. During the generation stage, the rules generated from the decoder will be firstly stored in a stack and converted to the tree with the first-in-last-out (FILO) principle.

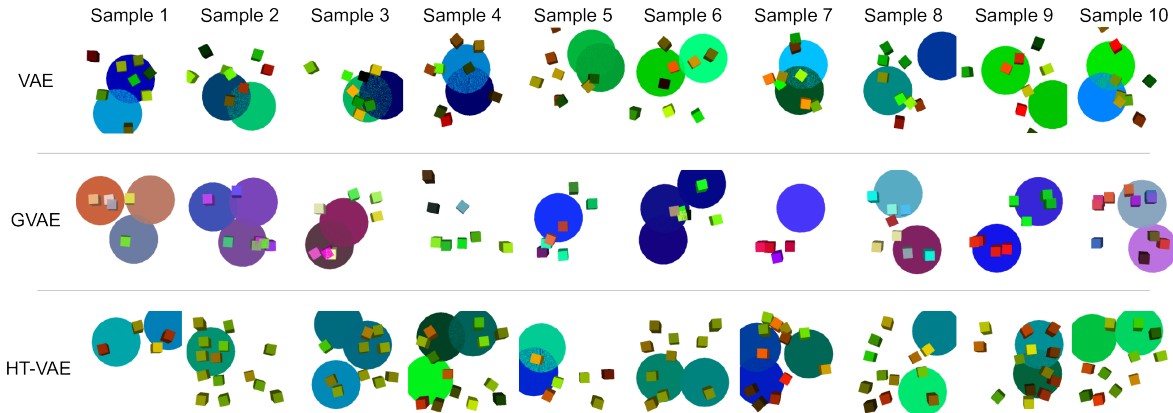

Figure 10: We randomly generate 10 samples by sampling in the latent space of VAE, GVAE, and T-VAE.

## A.5 Additional Qualitative Results

In Figure 10, we show samples randomly generated from VAE, GVAE, and T-VAE. The results show that all three models are able to generate diverse samples. Specifically, samples generated by VAE always have 2 plates and 8 boxes due to the fixed input data dimension. In contrast, samples from GVAE and T-VAE have variable numbers of plates and boxes.

In Figure 11, we show 4 more generated scenarios from the *Point Attack* methods with prediction results from 4 segmentation models. For better visualization, we also show three detailed figures. In Figure 12 and Figure 13, we show more results from *Pose Attack* and *Scene Attack* methods. Scenes generated by *Scene Attack* follow basic traffic rules.

## A.6 Segmentation Models in LiDAR Scene Generation

**PointNet++** This model Qi et al. (2017) directly uses point-wise features in the 3D space as the backbone to deal with the segmentation problem. Although this model does not have impressive results on the Semantic KITTI dataset, we select it because it influences a lot of existing point cloud processing models. We use the code from this repository and train the model on Semantic KITTI dataset by ourselves following the original training and testing split setting.

**PolarSeg** This model Zhang et al. (2020) converts the data representation from the 3D Cartesian coordinate to the Polar coordinate and extracts features with 2D convolution layers. We use the code from this repository and use the pre-trained model provided by the authors. Since we only consider the vehicle class, we change all other labels to non-vehicle class.

**SqueezeSegV3** This model Xu et al. (2020) projects 3D point clouds to 2D range maps and extracts features with 2D convolutions from the range maps. We use the code from this repository and use the pre-trained model provided by the authors. Since we only consider the vehicle class, we change all other labels to non-vehicle class.

**Cylinder3D** This model Zhou et al. (2020) converts the data representation from the 3D Cartesian coordinate to the Polar coordinate and divides the space into blocks with a cylinder representation. We use the code from this repository and use the pre-trained model provided by the authors. Since we only consider the vehicle class, we change all other labels to non-vehicle class.

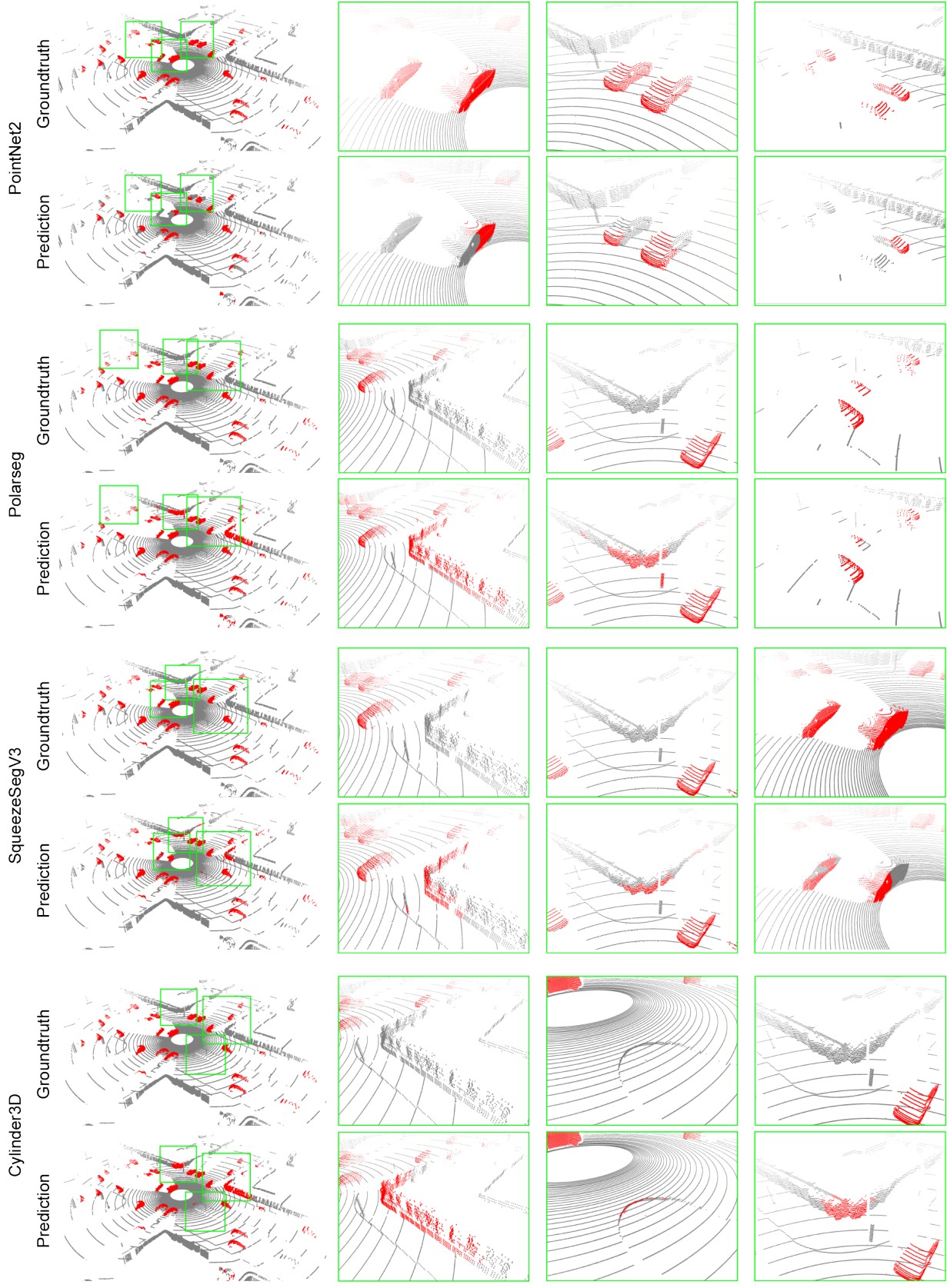

Figure 11: More results for *Point Attack* method with the *Intersection* background.

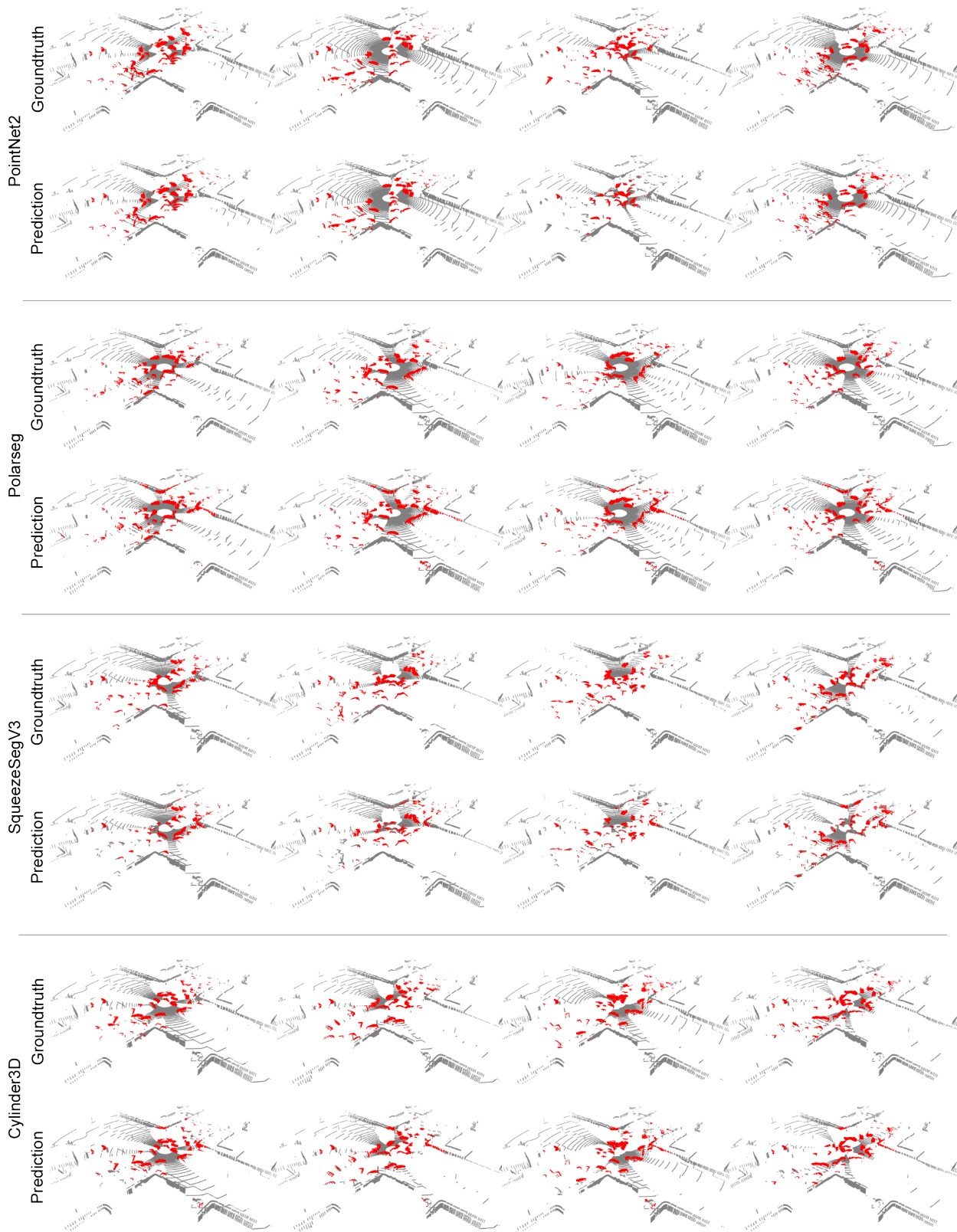

Figure 12: More results for *Pose Attack* method with the *Intersection* background.

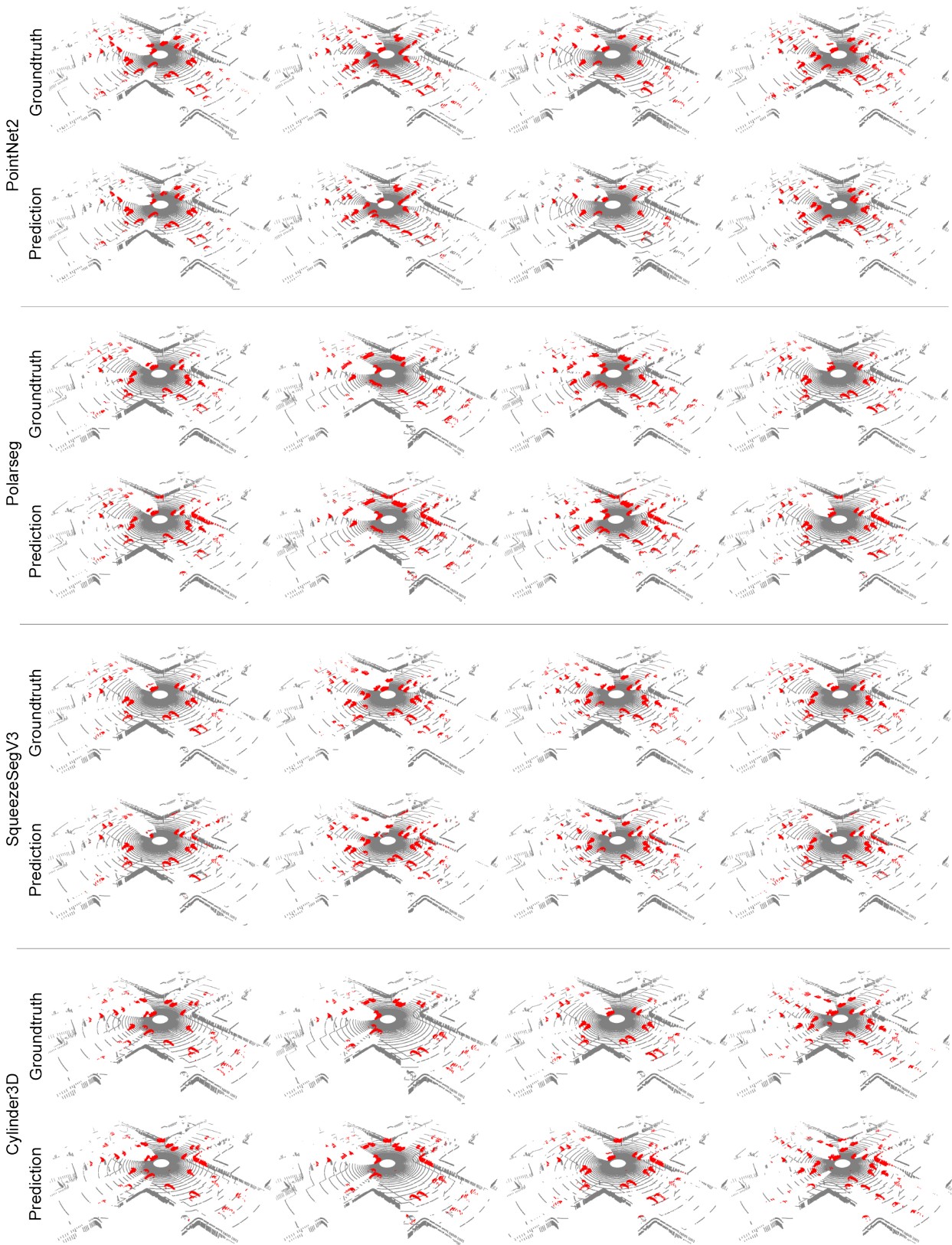

Figure 13: More results for *Scene Attack* method with the *Intersection* background.

