# OpenReview forum: "Semantically Adversarial Scene Generation with Explicit Knowledge Guidance for Autonomous Driving"
_TMLR — Rejected by TMLR_

### Review · Reviewer_N22U · 2022-11-06

**Summary Of Contributions:**

The paper proposes a semantically adversarial generative framework to sample adversarial images for training autonomous driving models. The procedure involves first finding a graph in which the nodes represent the property of the object and the edges corresponding to their relation. The tree information is then used in conjunction with a VAE model to construct semantically meaningful adversarial images. The proposed method is validated on a synthetic dataset of reconstructing object representations as well as a real-world dataset for autonomous driving.

**Audience:**

Yes

**Claims And Evidence:**

Yes

**Requested Changes:**

Please try to significantly improve the presentation of your work. The overall description of the method is quite unclear at many points. Please carefully explain how the knowledge graph is constructed (what kind of annotations it needs), what $f$ and $g$ features are and how they are extracted. Also, please clarify the function class and the functions in 3.2.1. The current state of the paper is quite confusing and unclear and, therefore not ready for publication in TMLR.

**Strengths And Weaknesses:**

### Strengths
Although I cannot comment on the novelty of generative modeling using tree structures, the proposed method tackles an interesting problem. Namely, creating realistic adversarial images for autonomous driving models. The proposed method has applications beyond autonomous driving such as medical imaging and fraud detection.
The illustrations are neatly done and provide a visual guide for understanding the techniques.

### Weaknesses
The paper mainly suffers from poor presentation. There are several typos, errors in the derivations, undefined variables or concepts, etc. I will thoroughly discuss each in the following.

##### There are several errors in the equations:
- Equation (1): Based on the definition, $\mathcal{L}$ of knowledge cannot be negative. Also, IIUC, the final approach in Section 3.2.2 does not solve a constrained optimization problem; it simply corresponds to the sum of two losses. Also, it is common to use a dummy variable inside the equation and in the argmax operator, instead of the solution itself, e.g., $x = \arg\max_{u} \\mathcal{L}_t(u)$.
- Equation (3): the definition implies that at each break, $w$ can only be split into two (i.e., $K_n = 2^n$). However, the description implies otherwise. I suggest using a better notation that is more comprehensive.
- Section 3.1.2 is super unclear. I was not able to follow the notation in Equation (5). $f$ and $g$ are defined on the fly, without providing a concrete example (are these features extracted from the network?). The $\theta_c$ parameter of the classifier is never defined. In the last line, $q$ and $p$ are referred to as model parameters, instead of distributions.
- In Figure 2, other types of nodes such as Plane Node and Box Node are never defined or referred to in the paper.
- In Section 3.2.1, $f$ is overused as a function belonging to $\mathcal{F}$ and the features in Section 3.1.2.
- Again, the description of the knowledge representation graph in Section 3.2.1 is quite unclear. What do $f_1$ and $f_2$ represent? Please provide more concrete examples of what these functions correspond to.

##### Typos and writing:
- Make sure to use a plural form after Author et al.: For instance in page 1, Bau et al. (2020) dissect…, Plumerault et al. (2020) interpolate…
- Page 2, second paragraph: we propose a Semantically Adversarial Generation (SAG) framework.
- There are several \citep (instead of \citet) errors in Page 3: Hu et al. 2016 distill, data driven networks in Mahmoudabadbozchelou (this is part of the text, so use \citet), Yang & Perdikaris (2018) restrict…
- Page 5: stories -> stores

---

### Review · Reviewer_aWQ1 · 2022-11-10

**Summary Of Contributions:**

The authors propose a method for constructing generative models for adversarial examples in a highly structured setting. They describe an tree-structured auto-encoder that encodes and decodes the scene hierarchically. This allows them to introduce domain knowledge in the form of constraints, which take the form of rules about nodes in the tree representation, or rules about the relationship between nodes. By incorporating this knowledge into the generation process, the authors generate more realistic adversarial examples that follow real-world rules and constraints.

**Audience:**

Yes

**Claims And Evidence:**

Yes

**Requested Changes:**

- More detail on exactly what is new in the structured autoencoder approach of the paper. The authors mention other tree-structured generative models in the section ‘structural deep generative models’ but it’s not clear how much of their approach is new and how much depends on these existing works.
- In `Section 3.2.1`, the authors describe the process of ‘knowledge projection.’ It seems like this is done by applying a series of rules. The example they give is ‘ if one node represents blue, its child nodes should represent red.’ However, it isn’t clear how we can ‘flip’ properties of certain nodes if they are encoded in the latent properties `g` of the node.
- There are a number of places in `Algorithm 2` where the authors write things like $x ← p(x | z, \theta)$, when I think they mean to sample $ x ~ p(x | z, \theta)$.
- In equation (1), $L_K$ is a ‘semantic loss’, which I interpret as evaluating how much the configuration `x` violates some set of rules K. In (12) it measures the difference between the generated tree and a version of the tree that has been modified to obey a set of rules. It’s not clear to me how these are the same loss.
- In Equation (13), the authors find $z$ by minimizing an expression containing $L_K(x, x^\prime)$. However, in `Algorithm 2`, they have $x^\prime = ApplyK(K, x)$ so it doesn’t seem like $L_K(x, x^\prime)$ depends on z. Did they instead mean to minimize $- \log(p(x^\prime | z^\prime, \theta)) + \frac{1}{2}||z - z^\prime||^2$? I might be misunderstanding some notation here (or the definition of L_K), but I think it would be helpful to clarify this proximal objective.
- `Figure 5 (b-c)` are difficult to interpret. Are these on the same scale?
- In `Synthetic Scene Reconstruction`, the task setup isn’t clear to me. Do the methods have access to the reconstruction loss $L_t(x) = ||S - R(x)||_2$ and its gradient? If so, how are they allowed to use it? In general, it seems like this task should be very easy for, e.g. a well-trained VAE.
- Using the `T-VAE` method without knowledge integration, how is the search performed? Since different `z` will lead to different tree structures, it doesn’t seem entirely differentiable. Is there a gradient estimation trick here, or do they use black box optimization?
- In `Section 4.2.2` the authors suggest that it typical not to have access to the parameters/gradients of the segmentation model. Aren’t these adversarial examples usually generated in order to further train the victim model? In that case, wouldn’t we have access to those gradients? If gradients are used instead of black box, how does T-VAE-SAG perform?
- In `Section 4.2.3: Adversarial Training`, I would like more details on the setup. Are these values the performance on adversarial examples with and without adversarial training? It would be nice to see performance on test data with and without adversarial training. Adversarial training seems like the most compelling application to me, so I would like some more detail here.

**Strengths And Weaknesses:**

## Strengths

- The authors motivate the problem well and give a clear summary of existing work.
- The paper has some great figures explaining the hierarchical structure of the T-VAE. These are really effective in my opinion.
- The application is quite compelling. Producing adversarial examples for autonomous driving systems could allow training on difficult situations without reproducing them on the road. Producing *********realistic********* adversarial examples is still a significant challenge that is addressed in this paper.

## Weaknesses

- I was a little unclear on exactly which parts of this method are new. Is the tree-structured VAE new? Or just the method of imparting structured knowledge?
- There are a few areas that I found confusing (detailed in the ‘Changes’ section). In particular, the definition of the loss function $L_K$ seemed inconsistent (or else I misunderstood it). The definition of the proximal operator used in `Algorithm 2` to optimize `z` was also unclear to me—details below. Since this algorithm is central to the paper, it’s important that it is very clear.
- The `Synthetic Scene Reconstruction` task seems trivial if the various approaches have access to the gradient of the reconstruction error. Perhaps I have just misunderstood the task. I think this section could use some clarification on exactly what the task is, what information each method has, and how they are trained.
- In the `Adversarial Driving Scenes Generation` task, it’s unclear why the authors say that parameters and gradients generally aren’t available for the segmentation models. I thought the adversarial examples were intended to improve training, in which case we would need these gradients. I would be interested to see these same results if gradients were made available to all methods.
- I am going to say that the claims *are* supported by evidence in the paper but I think this is marginal as it stands. I would really like to better understand the `Synthetic Scene Reconstruction` task (and exactly what information was available to, e.g., a VAE), as well as results on the test data after adversarial training in the `Adversarial Driving Scenes Generation` task.

---

### Review · Reviewer_1fnB · 2022-11-13

**Summary Of Contributions:**

This paper proposes a method to incorporate a predefined set of semantic (knowledge) constraints in adversarial scene generation. The key technique of the method is to build a tree-structured VAE to learn hierarchical scene representation, then impose the constraints to the corresponding tree structure of the generated scene. Experiments provide some evidence that when compared with existing methods, the adversarial scene generated by the proposed method satisfies the knowledge constraints better and transfer across different models better.

**Audience:**

Yes

**Claims And Evidence:**

No

**Requested Changes:**

1. Address my main concerns raised above

2. Define the threat model clearly before introducing your method (e.g., adversarial strength, what means for an adversarial attack to be successful)

3. Explain the definition of IoU.

4. The paper frequently uses terms like stick, box, plate, etc. They are difficult for readers to understand without specific knowledge. Consider make these terms consistent with clear definitions.

5. Explains the definition of L_k in Equation 12

**Strengths And Weaknesses:**

I like the motivation of imposing semantic constraints to the generated scene. This considers the feasibility of the generated adversarial images. For example, most-commonly encountered adversarial scene for an autonomous vehicle deployed in real world is more likely to be natural ones, which are semantically meaningful. Defining the underlying semantic constraint is challenging, but the paper does give some specific examples on how to define such semantic rules based on domain-specific knowledge. The proposed tree-structured method for incorporating the semantic constraints is new to me. It decomposes the generating process into two stages: it first generates a tree-structured VAE to represent the hierarchical relationship between objects for standard scene, then modify the decoder tree to satisfies the semantic constraints. However, I have some main concerns of the paper about the vague threat model and unfair evaluation procedure, which make me difficult to evaluate the contributions of the paper.

First, it is unclear to me regarding the attacker goals for the adversarial scene generation. The paper mentions several places in the paper that the goal is to lower the performance of the system, but this is very vague from a security standpoint. For example, in terms of your experiments on driving scenes generation, how do you define which adversarially generated scene is successful with respect to an image segmentation algorithm? In the experiments, you use Intersection over Union (IoU) as the metric for measuring the performance, but it is unclear to me how to translate an IoU score to attack successfulness. Also, there are no explanations on what IoU measures.

Second, there are no descriptions of the adversarial strength for modifying the scene. This is relevant to the first concern – since you do not clear specify the threat model, it is hard to see whether the comparisons between different type of adversarial scene generation methods are fair. For example, the adversarial distance is defined in the input space for Point Attack, but your method clearly modifies the latent space.

Other questions:

1. It seems that the proposed generating process heavily relies on the learned T-VAE in the first stage, but the knowledge constraints are only imposed in the second stage for generating an adversarial scene. Why not enforce the knowledge constraints for learning the T-VAE at the first place?

2. In general, what types of knowledge rules can be incorporated into the proposed framework?

3. Based on the knowledge rule, how do you define the loss of L_K for a generated scene? It is hard for me to parse the terms of its definition proposed in Equation 12.

---

### Review · Reviewer_1QES · 2022-11-14

**Summary Of Contributions:**

One of the problems of adversarial training is that the adversarially generated samples might not be possible in real life. To circumvent this problem, the authors propose to learn a hierarchical latent decomposition of scenes in order to impose some semantic structure in such hierarchy. Thanks to that, the authors are able to generate adversarial scenes that are more realistic and improve adversarial training. Their model is tested on a series of experiments on toy and synthetic and autonomous driving environments.

**Audience:**

Yes

**Broader Impact Concerns:**

I have no broader impact concerns with this work. If the authors would like to add a section, I suggest talking about the possibility of human biases permeating the semantic constraints that are put to the model. For example, given some gender detection, who decides what features should be present in each gender?



**Claims And Evidence:**

Yes

**Requested Changes:**

* I recommend reworking section 3 given my comments and those of the other reviewers.
* Could you make the captions more self-contained?
* Could you add a limitations section? You could talk about the limitations about using the tree, or the fact that you present a general algorithm that could be applied to multiple domains although you focus on autonomous driving.

**Strengths And Weaknesses:**

Strengths
=======
* The problem of generating semantically consistent samples is interesting
* The proposed T-VAE is sound and the optimization algorithm too.
* Some parts of the paper such as the introduction, the plots, related work, are well-written and polished.

Weaknesses
=========
* I found Section 3 difficult to read. For example, in eq 12, is $CE(\hat{c}_i, \hat{c}'_i)$ adversarial because $\hat{c}'_i$ is a different node type? Why is it written as CE instead of $\mathcal{L}_c$? How is adversarial training performed?
* It took some time for me to know how to interpret  $f_1(A) \rightarrow \forall j f_2(Bj)$
* The proposed idea makes sense for simple setups, however, there are complex setups for which it might not be possible to list all the semantic rules that have to be applied.
* The plots look nice but they are difficult to interpret. I encourage you to improve the captions to make them more self-contained. Particularly Figure 5 b and c. Also it was not clear to me how to read Figure 7.

---

### Decision · Action_Editors · 2023-01-06

**Recommendation:** Reject

**Comment:**

The paper was reviewed by four reviewers. A main criticism is that the presentation of the paper needs much improvement. Another criticism is that the goal of the adversarial attack is not clearly defined. A third criticism is that it is unclear whether the proposed method can work for complex scenes.

The authors did not provide rebuttals and revision. Thus the paper cannot be accepted in its current form.



**Audience:**

The paper should be interesting to researchers working on adversarial generation and autonomous driving.

**Claims And Evidence:**

This paper proposes a method for generating adversarial scenes for autonomous driving. Key ingredients of the proposed method include a tree-structured VAE and constraints based on explicit knowledge.

The claims of the paper are partially supported by the experiments.